# Characteristics of medication-induced xerostomia and effect of treatment

**Kayoko Ito** [1]*, **Naoko Izumi**[2], **Saori Funayama**[1], **Kaname Nohno**[3], **Kouji Katsura**[4], **Noboru Kaneko**[5], **Makoto Inoue** [1,6]

1 Oral Rehabilitation, Niigata University Medical and Dental Hospital, Niigata, Japan, 2 Medical Affairs, Internal Medicine, Pfizer Japan, Inc., Tokyo, Japan, 3 Division of Oral Science for Health Promotion, Faculty of Dentistry & Graduate School of Medical and Dental Sciences, Niigata University, Niigata, Japan, 4 Department of Oral Radiology, Niigata University Medical and Dental Hospital, Niigata, Japan, 5 Division of Preventive Dentistry, Faculty of Dentistry & Graduate School of Medical and Dental Sciences, Niigata University, Niigata, Japan, 6 Division of Dysphagia Rehabilitation, Faculty of Dentistry & Graduate School of Medical and Dental Sciences, Niigata University, Niigata, Japan

๏ These authors contributed equally to this work.
* k-ito@dent.niigata-u.ac.jp

## Abstract

### Objective

Side-effects of medications cause xerostomia. There have been cases where a medication has been discontinued owing to its severe side-effects. Therefore, the xerostomia must be treated to ensure that the primary disease is managed effectively. This study analyzed the actual status of patients with medication-induced xerostomia and investigates factors associated with its improvement.

### Methods

This study assessed 490 patients diagnosed with medication-induced xerostomia who had an unstimulated salivary flow of $\leq$0.1 mL/min and received treatment for xerostomia at a xerostomia clinic. Patient age, sex, medical history, medications used, disease duration of xerostomia, and psychological disorders were recorded. The anticholinergic burden was assessed using the Anticholinergic Cognitive Burden scale. The unstimulated salivary flow was measured by the spitting method. According to their symptoms and diagnoses, the patients were introduced to oral lubricants, instructed on how to perform massage, and prescribed Japanese herbal medicines, and sialogogues. Factors associated with the subjective improvement of xerostomia and objective changes in the salivary flow rate were recorded at six months.

### Results

Xerostomia improved in 338 patients (75.3%). The improvement rate was significantly lower in patients with psychiatric disorders (63.6%) (P = 0.009). The improvement rate decreased as more anticholinergics were used (P = 0.018). However, xerostomia improved in approximately 60% of patients receiving three or more anticholinergics. The unstimulated salivary flow increased significantly more in patients who reported an improvement of xerostomia

**Data Availability Statement:** All relevant data are within the manuscript and its Supporting information files.

**Funding:** This study was funded by Pfizer Japan, Inc. One of the co-author belongs to Pfizer Japan,

Inc. and she performed analysis and preparation of the manuscript.

**Competing interests:** This study was funded by Pfizer Japan, Inc. Naoko Izumi, one of the co-authors, is an employee of Pfizer Japan Inc. and has contributed to preparation of the study design, data analysis, decision to publish and preparation of the manuscript. However, Pfizer Japan Inc. did not provided support in the form of salaries for authors other than NI. Pfizer is a marketing authorization holder of fesoterodine, an anticholinergic medicine for overactive bladder and pediatric neurogenic bladder. These do not alter our adherence to PLOS ONE policies on sharing data and materials.

(0.033±0.053 mL/min) than in those who reported no improvement (0.013±0.02 mL/min) (P = 0.025).

## Conclusion

Xerostomia treatment improved oral dryness in 75.3% of patients receiving xerogenic medications in this study. If xerostomia due to side-effects of medications can be improved by treatment, it will greatly contribute to the quality of life of patients with xerogenic medications and may reduce the number of patients who discontinue medications.

## Introduction

Xerostomia refers to the subjective feeling of dry mouth, while hyposalivation refers to a low salivary flow rate [1]. The overall estimated prevalence of dry mouth was 22.0% (95%CI 17.0–26.0%) [2]. The reported prevalence of hyposalivation among older adults is 17%-47%; it varies depending on the approach and definition of hyposalivation used in the studies [3]. Patients with hyposalivation complain of symptoms—such as difficulties in eating, swallowing, speaking, halitosis, a chronic burning sensation, and altered taste perception—that significantly diminish the patients' quality of life (QOL) [4]. Furthermore, decreased salivary production can lead to oral mucosal *Candida* infection and increase the risk of dental caries [4]. Moreover, Ohara et al. reported an association between anorexia and hyposalivation [5]. Hyposalivation may also lead to malnutrition in older adults [6].

Hyposalivation may develop as a side-effect of using certain medications or may be caused by Sjögren's syndrome, psychological conditions, irradiation, and/or physiological changes [7, 8]. Up to 21% of patients seeking treatment at specialized xerostomia outpatient clinics have medication-induced hyposalivation [9]. In residential aged care, more than 95% of the population burden of dry mouth arises as a result of medication use [10]. It is well-known that the use of anticholinergics results in hyposalivation as saliva secretion is triggered via the stimulation of muscarinic receptors by acetylcholine [1]. Anticholinergics administered to treat overactive bladder cause xerostomia at a rate of 16.9%-53.7% [11–13]. Moreover, xerostomia accounts for about 40% of the reasons for discontinuation of anticholinergics due to adverse events in overactive bladder patients [14, 15]. Therefore, medication-induced xerostomia may reduce drug compliance and interfere with treatments for primary diseases.

The treatment of xerostomia includes drug therapy with sialogogues (such as pilocarpine hydrochloride and cevimeline hydrochloride hydrate), Japanese herbal medicines, oral lubricants, stimulation with gum or tablets, and changes or dose reductions of xerogenic medications [16–19]. While several studies have described therapeutic interventions for xerostomia caused by Sjögren's syndrome [20, 21] or radiotherapy for the head and neck region [22, 23], there have been few reports on medication-induced xerostomia [18, 24].

If xerostomia due to side-effects of medications can be improved by treatment, it will greatly contribute to the QOL of patients with xerogenic medications and may reduce the number of patients who discontinue medications. Accordingly, this study aimed to analyze the characteristics of patients with medication-induced xerostomia and the effect of treatment of xerostomia.

## Methods

This was a case series analysis of a large clinical convenience sample for which there was an intervention.

### Participants

Among 1,378 patients who visited the Xerostomia and Taste Disorder Clinic at Niigata University Medical and Dental Hospital with a chief complaint of xerostomia between August 2003 and December 2019, medication-induced xerostomia patients who had an unstimulated salivary flow rate of ≤0.1 mL/min and received treatment for xerostomia were included in this retrospective study. Medication-induced xerostomia was diagnosed in patients treated with xerogenic medications with package inserts warning of hyposalivation or xerostomia as a side-effect. Patients with missing data regarding medical history or medications were excluded from the study. The requirement for informed consent was waived because of the retrospective study design. An opt-out period was provided for eligible patients, and those who did not consent to participate in this study were also excluded. This study was conducted in accordance with the ethical principles of the Declaration of Helsinki and was approved by the Niigata University Ethics Committee (2020–0306).

### Patient data

Patient age, sex, medical history, medications used, and disease duration of xerostomia were extracted from medical records. Polypharmacy was defined as the use of six or more medications. Anticholinergic drugs were identified with reference to past Japanese study, taking into account four anticholinergic scales and drugs approved in Japan [25]. The Anticholinergic Cognitive Burden (ACB) scale was used to classify patients according to the degree of anticholinergic burden [26, 27]. The number of xerogenic medications was also counted for each patient.

The presence of psychological disorders was investigated using the Japanese version of the 30-item General Health Questionnaire (GHQ-30) [28]. The maximum score was 30; patients scoring seven or more points were regarded as having psychiatric symptoms [29].

### Measurement of the unstimulated saliva

The unstimulated salivary flow was measured via the 15-minute spitting method. Patients were instructed to spit out their saliva into a cup. The weight of the saliva was measured, and 1 g was considered to be equivalent to 1 mL [30]. Patients were categorized into three groups: having severe hyposalivation (<0.033 mL/min), moderate hyposalivation (0.033–0.066 mL/min), and mild hyposalivation (0.067–0.1 mL/min).

### Classification of xerostomia

Although we collected the patients diagnosed as medication-induced xerostomia, coexisting causes of the xerostomia were identified according to the Diagnosis Chart for Xerostomia [31]. Patients with a GHQ-30 score ≥7 points or those who simultaneously experienced psychological stress events and onset of xerostomia were diagnosed with psychological stress-induced xerostomia. Patients meeting the diagnostic criteria for Sjögren's syndrome were diagnosed as having xerostomia caused by Sjögren's syndrome. Patients with a history of radiotherapy for the head and neck regions were diagnosed as having radiation-induced xerostomia. Patients who reported mouth breathing were diagnosed as having evaporation-induced xerostomia.

Patients with metabolic diseases such as diabetes mellitus were diagnosed as having systemic disease-induced xerostomia.

## Treatment methods

All patients were administered over-the-counter oral lubricants [32] and instructed salivary gland massage [33]. Parotid gland massage was performed by placing the palm on the anteroinferior area of the auricle and gently moving the palm in a circle. Submandibular and sublingual gland massage was performed by gently pushing the inner edges of the mandibles upward with the thumbs.

Pilocarpine hydrochloride or cevimeline hydrochloride hydrate were prescribed to patients with Sjögren's syndrome or radiation-induced xerostomia. Japanese herbal medicines, for example Byakkokaninjinto, Ninjinyoeitou were also prescribed according to the patients' symptoms [17]. When changes in medications or dose reduction were preferable to improve xerostomia, we asked the prescribing physicians to modify the medication.

## Patient follow-up

The patients' treatment status at six months was categorized into three groups: treatment continuation, treatment completion, or treatment discontinuation. Treatment completion was defined as treatment termination after consent with the patient. It was considered to be treatment discontinuation if patients were lost to follow-up prior to treatment completion. Changes in the severity of xerostomia were assessed as three levels; the patients as improvement, no improvement, and exacerbation. The salivary flow rate at six months was measured when possible. If follow-up did not continue until six months, the data at the time of completion were treated as data at six months.

## Statistical analysis

Continuous variables are presented as median and range and categorical variables are presented as frequency. Cross-tabulations were performed with the chi-square test or Fisher's exact test for categorical variables, and the Cochran–Armitage test was used for categorical variables with three or more categories. Univariate analyses were performed to identify factors associated with the presence or absence of subjective improvement of xerostomia at six months. The no improvement group included patients who reported no improvement or exacerbation after six months, and the improvement group included patients who reported improvement after six months. Analyses were performed in all patients as well as in a subgroup of patients who received anticholinergics. Univariate logistic regression analyses were performed using xerogenic medications or ACB scale scores as the explanatory variable and the presence or absence of subjective improvement at six months as the objective variable. In the analysis by treatment method, the pattern of 10% or more of cases was analyzed.

The salivary flow rates at the initial visit and those after six months of treatment were tested for normality using the Shapiro-Wilk and analyzed using the Wilcoxon test. The Mann-Whitney test was performed to examine the association between the presence or absence of improvement in xerostomia and the unstimulated salivary flow. Effect size was calculated by the difference in mean score divided by the SD of the baseline score. All statistical analyses were performed using SAS version 9.4 statistical software (SAS Institute Inc., Cary, NC, USA). Statistical significance was set at $P < 0.05$.

## Results

### Patient characteristics

We included 490 of the 1,378 patients in this study on the basis of the selection criteria. The median age of the patients was 70 years (range: 17–89 years), and 68.4% of the patients were aged $\geq$65 years (Table 1 and S1 Table). Most patients (n = 415; 84.7%) were female. The

**Table 1. Patient characteristics.**

| | |
|---|---|
| **Age $\geq$ 65 years** | 335 (68.4) |
| **Sex (Female)** | 415 (84.7) |
| **Disease duration $\geq$ 19 months** | 244 (49.8) |
| **Medical history** | |
| Hypertension | 182 (37.1) |
| Dyslipidemia | 107 (21.8) |
| Gastrointestinal diseases | 87 (17.8) |
| Psychiatric disorders | 84 (17.1) |
| Cerebrovascular diseases | 50 (10.2) |
| **Medication** | |
| **Total number of medications $\geq$ 6 drugs** | 213 (43.5) |
| **Xerogenic medications $\geq$ 3 drugs** | 268 (54.7) |
| **Therapeutic category** | |
| Central nervous system drugs | 130 (26.5) |
| Cardiovascular drugs | 49 (10.0) |
| Gastrointestinal drugs | 38 (7.8) |
| Urological drugs | 17 (3.5) |
| Respiratory system drugs | 1 (0.2) |
| **Anticholinergics** | |
| 1 drug | 134 (55.4) |
| 2 drugs | 67 (27.7) |
| $\geq$ 3 drugs | 41 (16.9) |
| **Anticholinergic Cognitive Burden scale $\geq$ 3 points** | 77 (15.7) |
| **General Health Questionnaire ($\geq$ 7 points)** | 235 (48.0) |
| **Unstimulated salivary flow rate** | |
| Mild | 78 (15.9) |
| Moderate | 86 (17.6) |
| Severe | 326 (66.5) |
| **Diagnosis** | |
| Medication-induced xerostomia alone | 42 (8.6) |
| Concomitant stress-induced xerostomia | 334 (68.2) |
| Concomitant evaporation-induced xerostomia | 200 (40.8) |
| Concomitant Sjögren's syndrome | 91 (18.6) |
| **Treatment methods** | |
| SGM+OL | 229 (46.7) |
| SGM+OL+Japanese herbal medicine | 165 (33.7) |
| SGM+OL+sialogogues | 51 (10.4) |
| SGM+OL+Japanese herbal medicine+sialogogues | 24 (4.9) |
| SGM+OL+dose reduction/drug discontinuation | 18 (3.7) |

SGM: salivary gland massage OL: Oral lubricant.

Data are expressed as number (percentage).

median disease duration was 19 months (range: 0–360 months). The most common comorbidity was hypertension (n = 182; 37.1%), followed by dyslipidemia (n = 107 patients; 21.8%). The median number of medications used was five (range: 1–22 medications), and the median number of xerogenic medications used was three (range: 1–14 medications). The most commonly used xerogenic medications were central nervous system drugs (n = 130 patients; 26.5%). Any type of anticholinergics were administered to 242 patients (49.4%): 134 (55.4%) received one anticholinergic, 67 (27.7%) received two anticholinergics, and 41 (16.9%) received three or more anticholinergics. The median ACB scale score was 2 points (range: 1–10 points), and 77 patients (44.3%) had an ACB scale score ≥3 points.

At baseline, the median 15-minute unstimulated salivary flow was 0.013 mL/min (range: 0.0–0.1 mL/min), and 326 patients (66.5%) had severe hyposalivation. Forty-two patients (8.6%) were diagnosed with medication-induced xerostomia only, whereas the remaining patients (n = 448; 91.4%) were diagnosed with more than one type of xerostomia—334 patients (68.2%) were also diagnosed with psychological stress-induced xerostomia, 200 (40.8%) were also diagnosed with evaporation-induced xerostomia, and 91 (18.6%) were also diagnosed with Sjögren's syndrome. A total of 229 patients (46.7%) were treated with salivary gland massage and oral lubricants, and 165 patients (33.7%) were treated with Japanese herbal medicine, massage, and lubricants. Sialogogues (pilocarpine hydrochloride or cevimeline hydrochloride hydrate) were prescribed to 51 patients (10.4%). Dose reduction or drug discontinuation was achieved in two patients (0.4%). One patient changed the gastrointestinal drug and the other changed the urological drug. Neither was an anticholinergic drug.

## Factors associated with improvement of xerostomia

At six months, 297 patients (60.6%) were continuing treatment, 157 patients (32.1%) had completed treatment, and 36 (7.3%) had discontinued treatment. Data regarding the improvement of xerostomia were available for 449 patients at six months, including 338 patients (75.3%) who reported improvement, 109 who reported no improvement (24.3%), and two (0.4%) who reported exacerbation.

The improvement of xerostomia is shown in Table 2. The improvement rate was significantly lower in patients with psychiatric disorders (63.6%) (P = 0.009) as well as among those who received anticholinergics (60.3%) (P = 0.010). No statistically significant difference was observed between patients treated with and without anticholinergics. However, the incremental improvement decreased as the number of anticholinergics increased, 78.7% among patients receiving one anticholinergic, 70.8% among patients receiving two anticholinergics, and 59.5% among patients receiving three or more anticholinergics (P = 0.018). Among patients receiving anticholinergics, the improvement rate tended to be low in patients with a lower unstimulated salivary flow.

The improvement rate did not differ significantly according to the number of types of xerostomia diagnosed. The improvement rate was significantly lower in patients diagnosed with evaporation-induced xerostomia (70.3%) (P = 0.045) and Sjögren's syndrome (66.2%) (P = 0.043) than in patients without these types. The improvement rate did not differ between patients undergoing different treatment methods.

According to the logistic regression analysis, the use of an increased number of xerogenic medications was significantly associated with a lower improvement rate of xerostomia (P = 0.014) whereas the ACB scale score was not significantly associated with the improvement rate (P = 0.091).

Table 2. Subjective improvement in xerostomia.

| | | Total patients (n = 449) | | | Patients on anticholinergics (n = 224) | | |
|---|---|---|---|---|---|---|---|
| | | n | Improved | P | n | Improved | P |
| **Age** | | | | | | | |
| < 65 years | | 141 | 99 (70.2) | 0.092[a] | 76 | 47 (61.8) | 0.006[a] |
| ≥ 65 years | | 308 | 239 (77.6) | | 148 | 117 (79.1) | |
| **Sex** | | | | | | | |
| Male | | 68 | 52 (76.5) | 0.805[a] | 34 | 24 (70.6) | 0.707[a] |
| Female | | 381 | 286 (75.1) | | 190 | 140 (73.7) | |
| **Disease duration** | | | | | | | |
| ≤ 18 months | | 225 | 173 (76.9) | 0.528[a] | 120 | 89 (74.2) | 0.752[a] |
| ≥ 19 months | | 218 | 162 (74.3) | | 101 | 73 (72.3) | |
| **Medical history** | | | | | | | |
| Hypertension | Yes | 173 | 136 (78.6) | 0.195[a] | 91 | 70 (76.9) | 0.300[a] |
| | No | 276 | 202 (73.2) | | 133 | 94 (70.7) | |
| Dyslipidemia | Yes | 98 | 80 (81.6) | 0.099[a] | 45 | 40 (88.9) | 0.008[a] |
| | No | 351 | 258 (73.5) | | 179 | 124 (69.3) | |
| Gastrointestinal diseases | Yes | 82 | 64 (78.0) | 0.52[a] | 40 | 29 (72.5) | 0.910[a] |
| | No | 367 | 274 (74.7) | | 184 | 135 (73.4) | |
| Psychiatric disorders | Yes | 77 | 49 (63.6) | 0.009[a] | 58 | 35 (60.3) | 0.010[a] |
| | No | 372 | 289 (77.7) | | 166 | 129 (77.7) | |
| Cerebrovascular diseases | Yes | 47 | 36 (76.6) | 0.825[a] | 29 | 20 (69.0) | 0.580[a] |
| | No | 402 | 302 (75.1) | | 195 | 144 (73.8) | |
| **Medication** | | | | | | | |
| **Number of medications** | | | | | | | |
| ≤ 5 drugs | | 197 | 141 (71.6) | 0.108[a] | 139 | 98 (70.5) | 0.241[a] |
| ≥ 6 drugs | | 252 | 197 (78.2) | | 85 | 66 (77.6) | |
| **Xerogenic medications** | | | | | | | |
| < 3 drugs | | 203 | 161 (79.3) | 0.072[a] | 52 | 42 (80.8) | 0.160[a] |
| ≥ 3 drugs | | 246 | 177 (72.0) | | 172 | 122 (70.9) | |
| **Anticholinergics** | | | | | | | |
| Yes | | 224 | 164 (73.2) | 0.312[a] | — | — | — |
| No | | 225 | 174 (77.3) | | — | — | |
| 1 drug | | — | — | — | 122 | 96 (78.7) | 0.018[b] |
| 2 drugs | | — | — | | 65 | 46 (70.8) | |
| ≥ 3 drugs | | — | — | | 37 | 22 (59.5) | |
| **Degree of anticholinergic burden** | | | | | | | |
| < 3 points on ACB scale | | — | — | — | 88 | 68 (77.3) | 0.557[a] |
| ≥ 3 points on ACB scale | | — | — | | 71 | 52 (73.2) | |
| **GHQ** | | | | | | | |
| ≤ 6 points | | 139 | 107 (77.0) | 0.366[a] | 65 | 51 (78.5) | 0.168[a] |
| ≥ 7 points | | 216 | 157 (72.7) | | 109 | 75 (68.8) | |
| **Unstimulated salivary flow rate** | | | | | | | |
| Mild | | 71 | 58 (81.7) | 0.131[b] | 36 | 31 (86.1) | 0.052[b] |
| Moderate | | 78 | 60 (76.9) | | 39 | 29 (74.4) | |
| Severe | | 300 | 220 (73.3) | | 149 | 104 (69.8) | |
| **Diagnosis** | | | | | | | |
| **The number of diagnosed types** | | | | | | | |
| Medication-induced xerostomia alone | | 40 | 32 (80.0) | 0.468[a] | 24 | 19 (79.2) | 0.486[a] |
| Multiple types | | 409 | 306 (74.8) | | 200 | 145 (72.5) | |

(*Continued*)

**Table 2.** (Continued)

| | | Total patients (n = 449) | | | Patients on anticholinergics (n = 224) | | |
|---|---|---|---|---|---|---|---|
| | | n | Improved | P | n | Improved | P |
| **Stress-induced xerostomia** | Yes | 306 | 225 (73.5) | 0.209[a] | 155 | 111 (71.6) | 0.417[a] |
| | No | 143 | 113(79.0) | | 69 | 53(76.8) | |
| **Evaporation-induced xerostomia** | Yes | 182 | 128 (70.3) | 0.045[a] | 98 | 67 (68.4) | 0.149[a] |
| | No | 267 | 210 (78.7) | | 126 | 97 (77.0) | |
| **Sjögren's syndrome** | Yes | 77 | 51 (66.2) | 0.043[a] | 32 | 20 (62.5) | 0.139[a] |
| | No | 372 | 287 (77.2) | | 192 | 144 (75.0) | |
| **Treatment methods [c]** | | | | | | | |
| SGM+OL | | 211 | 170 (80.6) | 0.155[b] | 116 | 89 (76.7) | 0.437[b] |
| SGM+OL+Japanese herbal medicine | | 154 | 111 (72.1) | | 70 | 48 (68.6) | |
| SGM+OL+sialogogues | | 43 | 32 (74.4) | | 18 | 14 (77.8) | |

[a]Pearson χ2 test.

[b]Cochran-Armitage test.

[c]Analyzed for patterns of more than 10% of cases.

SS: Sjögren's syndrome.

SGM: salivary gland massage.

OL: Oral lubricant.

Data are expressed as median (range).

### Factors associated with the salivary flow rate at six months

Overall, the salivary flow rate was significantly higher than at baseline in the 91 patients in whom the unstimulated salivary flow was measured at six months (Table 3). The effective size was grater in the patient whose unstimulated salivary flow rate was mild.

### Xerostomia and the unstimulated salivary flow at six months

Of 91 patients in whom the unstimulated salivary flow was measured at six months, 74 (81.3%) reported improved xerostomia and 17 (18.7%) reported no improvement. The mean increase of unstimulated salivary flow was 0.033±0.053 mL/min in patients who reported xerostomia improvement and 0.013±0.02 mL/min in those who reported no improvement in xerostomia (P = 0.025). A total of 57 patients who reported improvement (77.0%) and 9 patients who reported no improvement (52.9%) had increased salivary flow rates at six months.

## Discussion

This is the first study in which factors associated with the improvement of xerostomia and changes in the unstimulated salivary flow at six months were analyzed in patients with medication-induced xerostomia. Xerostomia improved in 59.5% of patients even if the patient who receiving three or more anticholinergics. This finding will contribute to physicians concerned with the patients who complaint with the side-effect of medications.

One of the treatment methods for patients with medication-induced xerostomia is replacing the medication causing hyposalivation or reducing the dose of the causative medication [24]. Although a previous study indicated that improvement was achieved in 41% of patients who changed medications [34], it is difficult to switch medications in actual clinical practice. In this study, dose reduction or drug discontinuation of the causative medication was achieved in only two patients (0.4%). However, the xerostomia was improved in 338 patients (75.3%),

**Table 3. Salivary flow rates.**

| | | n | Before treatment | At 6 months | P | effect sizes |
|---|---|---|---|---|---|---|
| **Age** | | | | | | |
| < 65 years | | 30 | 0.37±0.45 | 0.82±0.86 | <0.001 | 1.00 |
| ≥ 65 years | | 63 | 0.42±0.47 | 0.94±0.99 | <0.001 | 1.11 |
| **Sex** | | | | | | |
| Male | | 17 | 0.56±0.50 | 1.31±0.92 | <0.001 | 1.50 |
| Female | | 76 | 0.37±0.45 | 0.81±0.93 | <0.001 | 0.98 |
| **Disease duration** | | | | | | |
| ≤ 18 months | | 53 | 0.41±0.45 | 1.13±1.04 | <0.001 | 1.60 |
| ≥ 19 months | | 37 | 0.43±0.49 | 0.63±0.73 | 0.001 | 0.41 |
| **Medical history** | | | | | | |
| Hypertension | Yes | 43 | 0.47±0.47 | 1.04±1.06 | <0.001 | 1.21 |
| | No | 50 | 0.34±0.46 | 0.78±0.83 | <0.001 | 0.96 |
| Dyslipidemia | Yes | 26 | 0.45±0.45 | 1.33±1.16 | <0.001 | 1.96 |
| | No | 67 | 0.38±0.47 | 0.74±0.80 | <0.001 | 0.77 |
| Gastrointestinal diseases | Yes | 12 | 0.39±0.52 | 0.81±0.89 | 0.012 | 0.81 |
| | No | 81 | 0.40±0.46 | 0.91±0.96 | <0.001 | 1.11 |
| Psychiatric disorders | Yes | 19 | 0.32±0.45 | 0.82±0.82 | <0.001 | 1.11 |
| | No | 74 | 0.42±0.47 | 0.92±0.98 | <0.001 | 1.06 |
| Cerebrovascular diseases | Yes | 9 | 0.54±0.55 | 0.71±0.91 | 0.297 | 0.31 |
| | No | 84 | 0.39±0.46 | 0.92±0.95 | <0.001 | 1.15 |
| **Medication** | | | | | | |
| **The number of medications** | | | | | | |
| ≤ 5 drugs | | 51 | 0.37±0.45 | 0.80±0.75 | <0.001 | 0.96 |
| ≥ 6 drugs | | 42 | 0.44±0.48 | 1.02±1.15 | <0.001 | 1.21 |
| **Xerogenic medications** | | | | | | |
| < 3 drugs | | 42 | 0.39±0.47 | 0.78±0.78 | <0.001 | 0.83 |
| ≥ 3 drugs | | 51 | 0.42±0.47 | 1.00±1.06 | <0.001 | 1.23 |
| **Anticholinergics** | | | | | | |
| Yes | | 48 | 0.42±0.48 | 0.94±1.06 | <0.001 | 1.08 |
| No | | 45 | 0.38±0.46 | 0.86±0.82 | <0.001 | 1.04 |
| 1 drug | | 26 | 0.47±0.52 | 1.01±1.17 | 0.003 | 1.04 |
| 2 drugs | | 17 | 0.34±0.41 | 0.89±1.05 | 0.012 | 1.34 |
| ≥3 drugs | | 5 | 0.48±0.54 | 0.74±0.52 | 0.155 | 0.48 |
| **Degree of anticholinergic burden** | | | | | | |
| < 3 points on ACB scale | | 18 | 0.26±0.29 | 0.68±0.88 | 0.009 | 1.45 |
| ≥ 3 points on ACB scale | | 16 | 0.41±0.53 | 1.17±1.08 | 0.002 | 1.43 |
| **GHQ** | | | | | | |
| ≤ 6 points | | 38 | 0.33±0.42 | 0.71±0.69 | <0.001 | 0.90 |
| ≥ 7 points | | 55 | 0.46±0.49 | 1.03±1.08 | <0.001 | 1.16 |
| **Unstimulated salivary flow rate** | | | | | | |
| Mild | | 18 | 1.20±0.14 | 1.95±1.06 | 0.009 | 5.36 |
| Moderate | | 14 | 0.69±0.14 | 1.01±0.74 | 0.108 | 2.29 |
| Severe | | 61 | 0.10±0.14 | 0.57±0.70 | <0.001 | 3.36 |
| **Diagnosis** | | | | | | |
| Medication-induced xerostomia alone | | 6 | 0.40±0.52 | 0.92±0.63 | 0.006 | 1.00 |
| Concomitant stress-induced xerostomia | | 70 | 0.42±0.47 | 1.00±1.02 | <0.001 | 1.23 |
| Concomitant evaporation-induced xerostomia | | 45 | 0.40±0.47 | 0.94±0.99 | <0.001 | 1.15 |

*(Continued)*

**Table 3.** (Continued)

|  | n | Before treatment | At 6 months | P | effect sizes |
|---|---|---|---|---|---|
| Concomitant Sjögren's syndrome | 18 | 0.20±0.37 | 0.33±0.52 | 0.036 | 0.35 |
| **Treatment methods**[a] |  |  |  |  |  |
| SGM+OL | 21 | 0.43±0.51 | 1.09±0.98 | 0.005 | 1.29 |
| SGM+OL+Japanese herbal medicine | 51 | 0.45±0.46 | 0.99±1.02 | <0.001 | 1.17 |
| SGM+OL+sialogogues | 7 | 0.44±0.59 | 0.70±0.91 | 0.137 | 0.44 |

[a]Analyzed for patterns of more than 10% of cases.

SGM: salivary gland massage.

OL: Oral lubricant.

Data are expressed as mean±SD.

indicating that xerostomia can be alleviated via treatment in patients receiving medications that may cause xerostomia. Therefore, reduced drug compliance due to xerostomia may be preventable.

There are various pathogenic mechanisms of medication-induced xerostomia. Saliva secretion is controlled by the autonomic nervous system, whereas fluid secretion is controlled by the parasympathetic nervous system [1]. Mechanisms of hyposalivation drug action may involve drug interference with transmission at the parasympathetic neuroeffector junction, actions at the adrenergic neuroeffector junction, or the depression of central connections of the autonomic nervous system [35]. Diuretics may affect the movement of water and/or electrolytes through the cell membrane of salivary acinar cells [35]. Antihypertensive drugs can cause xerostomia due to their effects on the regulation of calcium, which has an essential role in saliva secretion [36]. Tricyclic antidepressants can cause the inhibition of cholinergic, histaminic, and α1 adrenergic receptors, resulting in xerostomia [36]. Thus, the mechanisms of hyposalivation are dependent on the causative medication. The incidence of hyposalivation also differs based on pharmacokinetic (such as drug dose, absorption, and interactions) and physiologic (such as age, sex, and body weight) factors. In this study, 8.6% of patients were diagnosed with only medication-induced xerostomia, and 91.4% were diagnosed with multiple types of xerostomia. As xerostomia is rarely caused by medications alone, it is very difficult to interpret.

In this study, the improvement of xerostomia was not significantly different between patients with and without polypharmacy. The definition of polypharmacy varies; it is defined as the use of two or more medications in some studies and as the use of 11 or more medications in others [37]. In this study, polypharmacy was defined as the use of six or more medications [38]. The findings of this study suggest that the more number of taking xerogenic medications are, the more difficult it is to improve xerostomia. Effort to reduce the number of xerogenic medications may lead to benefit to patients with xerostomia.

The most common therapeutic intervention in this study was the combination of salivary gland massage and oral lubricants. There were no significant differences in subjective improvement rates by additional treatments, namely Japanese herbal medicines or sialogogue. In Japan, sialogogues are approved to treat Sjögren's syndrome or xerostomia associated with radiotherapy. In both diseases, the salivary glands are pathologically damaged [8], the salivary flow rate improvement may not have been achieved despite treatment. This theory is consistent with the finding that the improvement of xerostomia was significantly lower in patients with concomitant Sjögren's syndrome causing xerostomia than in those without Sjögren's syndrome. However, the treatment selection was biased in this study, as this was an observational

study. Treatment methods were selected in based on the severity of xerostomia and at the requests of the patients. Thus, it is possible that drug therapy was selected only for patients with severe xerostomia, and that patients with mild xerostomia were treated only with massage and oral lubricants. Despite this limitation, the findings of this study suggest that treatment improves medication-induced xerostomia in 75.3% of patients. Previous studies have reported the effects of pilocarpine preparations [39, 40] and oral lubricants [41] for patients with medication-induced xerostomia. Because in these reports, only the changes in the salivary flow rate were evaluated, improvements in xerostomia are unclear in these previous studies. Although the degree of improvement was classified into three categories in this study, detailed studies using the visual analog scale and the Facial scale are necessary.

Although the salivary flow rate increased significantly more in patients who reported improvement of xerostomia than in those who reported no improvement of xerostomia, there was a large variation of increase rate. The salivary flow rate did not increase in 23.0% of patients who reported improved xerostomia. In contrast, 52.9% of patients who did not report improved xerostomia had increased salivary flow rates. The inconsistency between the xerostomia and the objective index of the salivary flow rate has been previously reported [42]. Due to these inconsistent findings, it is unclear how much the salivary flow rate must increase to improve xerostomia. The appropriate threshold to measure improvement in xerostomia is also unclear. In this study, any increase from the baseline salivary flow rate was regarded as an increase; however, the criteria for improvement should be examined in the future.

This study has several limitations. First, only the ACB scale was used to evaluate the anticholinergic burden in this study. The scales to measure the burden of anticholinergic effects include the Anticholinergic Risk Scale (ARS) [43], anticholinergic drug scale (ADS) [44], and the ACB scale. The ARS is applicable for 49 medications, and the ADS is not applicable to fesoterodine, propiverine, and solifenacin, which are therapeutic drugs commonly used for overactive bladder in Japan. Therefore, this study used the ACB scale, which was developed by Boustani et al. [45] to measure the accumulative anticholinergic cognitive burden resulting from the total medications taken by older adults. As this scale was not developed in Japan, the applicable medications may not encompass the anticholinergics administered to patients in this study. In addition, it is necessary to search the interaction of medications. Second, the number of patients who underwent measurement of the salivary flow rate at six months was small. In this study, unstimulated salivary flow was measured for 15 minutes according to the diagnostic criteria for Sjögren's syndrome. Many patients had systemic diseases and had consultations in multiple clinical departments in our hospital, including internal medicine, urology, and orthopedic surgery, within a few hours. Therefore, it was difficult to set aside enough time to conduct salivary flow tests for all patients. There is an oral moisture-checking device called Mucus®, which differs from unstimulated salivary flow [46]. With this device, oral mucosal moisture can be measured within a few seconds. In future studies, the use of other methods of objective assessment should be considered.

## Conclusion

This research indicated that 75.3% of patients treated with xerogenic medications reported improvement in xerostomia after six months of treatment with salivary gland massage and oral lubricants, and others. Although the improvement rate was lower among patients who were administered more anticholinergics, approximately 60% of patients who received three or more anticholinergics reported improvement. The doctors who prescribe xerogenic medications should attempt to manage xerostomia. This would prevent patients with xerostomia from no longer taking medicine for the primary disease.

## Supporting information

**S1 Table. Database.**
(XLSX)

## Acknowledgments

We would like to express our appreciation to Shinichi Kanazawa at A2 Healthcare Corporation, who was involved in the data analyses and to Goki Jitsukata and Yuya Kanauchi at Pfizer Inc., who assisted in the composition of the discussion section.

## Author Contributions

**Conceptualization:** Kayoko Ito, Naoko Izumi, Saori Funayama, Kaname Nohno, Kouji Katsura, Noboru Kaneko, Makoto Inoue.

**Data curation:** Kayoko Ito, Saori Funayama, Kouji Katsura, Noboru Kaneko, Makoto Inoue.

**Formal analysis:** Kayoko Ito, Naoko Izumi, Kaname Nohno, Noboru Kaneko.

**Funding acquisition:** Kayoko Ito, Naoko Izumi.

**Investigation:** Kayoko Ito, Saori Funayama, Kaname Nohno, Kouji Katsura, Noboru Kaneko.

**Methodology:** Kayoko Ito, Naoko Izumi, Kaname Nohno.

**Project administration:** Kayoko Ito.

**Resources:** Kayoko Ito, Naoko Izumi.

**Supervision:** Kayoko Ito, Makoto Inoue.

**Validation:** Kayoko Ito, Kaname Nohno.

**Visualization:** Kayoko Ito.

**Writing – original draft:** Kayoko Ito.

**Writing – review & editing:** Kayoko Ito, Naoko Izumi, Saori Funayama, Kaname Nohno, Kouji Katsura, Noboru Kaneko, Makoto Inoue.

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
