## [Decision Letter · Decision Letter 0]

26 Oct 2022

PONE-D-22-09854Characteristics of medication-induced xerostomia and factors associated with treatment responsePLOS ONE

Dear Dr. Ito,

Thank you for submitting your manuscript to PLOS ONE. After careful consideration, we feel that it has merit but does not fully meet PLOS ONE’s publication criteria as it currently stands. Therefore, we invite you to submit a revised version of the manuscript that addresses the points raised during the review process.

There are some minor concerns on the details of the methodology and results. However, a reviewer also raised a concern about conflict of interest. Please review the comments and address them accordingly.

We look forward to receiving your revised manuscript.

Kind regards,

Sompop Bencharit, DDS, MS, PhD, FACP

Academic Editor

PLOS ONE

Journal Requirements:

2. Thank you for stating the following in the Competing Interests:

“This study was funded by Pfizer Japan, Inc..

Naoko Izumi was involved in research and preparation of the manuscript as an employee of Pfizer Japan Inc.”

We note that one or more of the authors have an affiliation to the commercial funders of this research study : Pfizer Japan, Inc.

Reviewers' comments:

Reviewer's Responses to Questions

**Comments to the Author**

1. Is the manuscript technically sound, and do the data support the conclusions?

Reviewer #1: Partly

Reviewer #2: Yes

2. Has the statistical analysis been performed appropriately and rigorously? 

Reviewer #1: Yes

Reviewer #2: Yes

3. Have the authors made all data underlying the findings in their manuscript fully available?

Reviewer #1: Yes

Reviewer #2: Yes

4. Is the manuscript presented in an intelligible fashion and written in standard English?

Reviewer #1: Yes

Reviewer #2: Yes

5. Review Comments to the Author

Reviewer #1: General comments

The data are interesting, but the authors need to do a better job of presenting and describing their findings, and of making the case for conducting the study (Introduction section).

They have used something like triple spacing – which is not reader-friendly at all. I suggest that they use 1.5 spacing, which is much easier to read.

I am somewhat concerned by one of the authors being an employee of the private-sector funder (Pfizer). There needs to be a very careful explanation of how any associated conflicts of interest were managed in respect of this research.

The authors should be very clear about when they are referring to “xerostomia” (the symptoms of dry mouth) and when they are referring to low salivary flow (or salivary gland hypofunction, SGH). The subjective and objective aspects of dry mouth do not necessarily coincide, and the authors’ choice of terms should reflect that dichotomy. People with an unstimulated flow rate of <0.1 mL/min do not necessarily have xerostomia, but they definitely have SGH.

The study design is not clear, but I think I have worked it out – it could be described as a retrospective cohort study of a large clinical convenience sample, but there is an actual intervention, so it’s not strictly observational (and therefor enot a cohort study as such). I think it is a case series analysis a large clinical convenience sample for which there was an intervention (or interventions). The authors need to be very clear about their design, because the reader’s understanding of the findings is very much dependent on the design being explicit.

Always hyphenate “side-effect”. It is a compound word.

Headings and subheadings are huge – do we really need them that big?

Section comments

Title

This needs to better reflect the study design and the research question, and the term “xerostomia” should be replaced by “dry mouth”.

Abstract

Will need rewriting anyway, but some comments follow.

The logic of the first two sentences escapes me.

Use the past tense in describing findings. Do not rely on P vlues – put some actual data in the Results paragraph. Your study should not be a P value hunt – see Amrhein et al, Nature 2019; 567: 305-307.

Conclusion – that phrase “…can prevent decreased drug compliance for the primary disease” is also a mystery – what are you saying? By what logic?

Introduction

Overall, this should make a more compelling case for conducting the study.

Paragraph 1

Sentence 1 – change “decreased” to “low”. For a better and more recent estimation of prevalence rates, see Agostini BA et al. How common is dry mouth? Systematic review and meta-regression analysis of prevalence estimates. Brazilian Dental Journal 29: 606-618 (2018).

Last sentence – that is an unsupported sweeping statement – where is the evidence for that assertion?

Paragraph 2

Sentence 2 – medication-induced dry mouth accounts for well over 95% of cases of dry mouth – see the recent US Surgeon General’s Report on Oral Health.

Sentence 3 – That Sreebny list is far too inclusive, and you should avoid using those specific numbers.

Paragraph 3

See the Surgeon General report for a good overview of the therapeutic approaches for treating dry mouth – there is a very useful Table there. Last sentence of para 3 – change the awful “regarding” to “on”.

Paragraph 4

Sentence 1 – “…and eventual imrpovement of the primary disease” – really? That’s drawing a very long bow. How might that work? Next sentence – change “Therefore” to “Accordingly”. And specify the research question. The way the last sentence is worded is far too loose and woolly, and the reader has no idea of what the study is about.

Methods

Lines 93-94 – patients were not identified; the drugs were. Rewrite that sentence accordingly.

Lines 105-107 – express those as mL/min, not per 15 min.

Lines 111 to 113 – how do you know they “developed xerostomia”? Or did they “have” the condition?

Line 121 – “educated regarding” is poor wording – they were “made aware of” it, or taught how to do it – something like that is better.

Lines 126-7 – what herbal medicines? How given? When? Why? No reference – just a throwaway comment is given.

Line 135 – how exactly was xerostomia assessed? What was asked, and what were the response options?

Line 137 should be reworded.

Statistical analysis section – concentrate less onf the test stistics (such as chi-square tests) and more on the procedures – such as “cross-tabulations”, etc etc.

Results

Line 159 makes no sense at all. Line 166 also has problems – “all types” – really? Do you mean “any type”?

Table 1 is just a list. What not make it more informative for readers by cross-tabulating by sex or age group? And no Table should continue onto a second page. Think about reducing your use of gridlines – they actually distract the eye from the data.

In the Results text, do not repeat Table data – the reader can see the data in the Tables; your job is to draw his/her attention to the important parts and features. Have one paragraph of Results text per Table – this helps the reader to navigate your paper – and introduce each Table at the beginning of its paragraph, not at the end. Concentrate on what the data show, NOT on the P values – they are far less important.

Avoid using the term "compared to" when making comparisons - use 'than' – for example, in Line 227, replace “increased compared to baseline” with “was higher than at baseline’.

Table 3 would be enhanced by presenting effect sizes (the difference in mean score divided by the SD of the baseline score, and prsented to 1 decimal place. For example, that for those <65 years is 1.0, which is a large effect. And you will be able to demonstrate that the ES for the Sjogren’s patients was lower (0.4) than for those with medication-induced xerostomia (1.0) – that is important information, and much more useful and informative than the P values – which you could indicate with a footnote anyway.

Discussion

This will need rewriting anyway – in its current form, it is too long and discursive. In the Discussion and conclusion, use the term 'findings' rather than 'results'. See Docherty and Smith, BMJ 1999; 318: 1224-5 for how to structure a Discussion section. It is a useful structure. As a general rule, the first paragraph of the Discussion should briefly reiterate what the study did and what it showed. The second paragraph should address the weaknesses of the study design and measures, etc. The paragraphs which follow should then discuss how the findings support or refute the current literature. The final paragraph should tie it all together – so what? Where next? What are the implications for practice?

The current conclusion is not useful – what’s the take-home message from this study?

Reference 23 – is that legit?

Reviewer #2: Dear authors,

Congratulations for the study. It will definitely contribute to the scientific field of Pharmacology. Though I would suggest small changes to improve the quality of the reported results.

Line 63 - You better explain what OAB stands for. Does it refer to overreactive bladder?

Line 72 - You better explain what QOL stands for. Does it refer to quality of life?

In the results section I would suggest you to diplay the results in tables only as the majority of them were explained in the text and repeated on the tables. You could have a small paragraph describing a little about what will be displayed without mentioning the figures. I would also suggest you to re-organize table 1 as data are not clear. For instance , I took some time to understand that data presented as 70 on line 2, stands for the median age, and on the next line the figure stands for the size of the sample >= 65 years old. Its seems confuse!

On the headline of the columns write to what the figure stands for and in brackets you could write frequency in %. You better verify the total percentage for medical history.

Line 183 - What kind of Japanese Herbal medicine was used? Do they all have the same mechanism of action?

Did the use of anticholinergic drugs present any type of pharmacological interaction with the other drugs in use by patients? Did you have this data on patient´s record? Nothing was mentioned on the text about it.

Best regards.

6. PLOS authors have the option to publish the peer review history of their article (what does this mean?). If published, this will include your full peer review and any attached files.

Reviewer #1: No

Reviewer #2: No

---

## [Author Response · Author response to Decision Letter 0]

12 Dec 2022

Reviewers: 

We thank the reviewers for the helpful comments on our manuscript. In accordance with the reviewers’ suggestions, we have carefully revised the manuscript, and we consider that it has been greatly improved as a result. Our responses to the reviewers’ comments are indicated below. All of the revised sections have been highlighted in blue.

Editor

We have modified our file to meet PLOS ONE’s style.

2. Thank you for stating the following in the Competing Interests:

"This study was funded by Pfizer Japan, Inc..　Naoko Izumi was involved in research and preparation of the manuscript as an employee of Pfizer Japan Inc."　We note that one or more of the authors have an affiliation to the commercial funders of this research study : Pfizer Japan, Inc.

This study was funded by Pfizer Japan, Inc. and Naoko Izumi, who is one of the co-authors, is an employee of Pfizer Japan Inc. and has contributed to preparation of the study design, data analysis, decision to publish and preparation of the manuscript. However, Pfizer Japan Inc. did not provide support in the form of salaries for authors other than Naoko Izumi. Pfizer is a marketing authorization holder of fesoterodine, which is an anti-cholinergic medicine for overactive bladder and pediatric neurogenic bladder. This information does not alter our adherence to PLOS ONE policies on sharing data and materials.

We have added the Supporting Information to the end of our manuscript.

Reviewer: 1

General comments:

1) Reviewer’s comment 

They have used something like triple spacing - which is not reader-friendly at all. I suggest that they use 1.5 spacing, which is much easier to read.

We thank the reviewer for the advice. We have changed the spacing from 2 to 1.5.

2.I am somewhat concerned by one of the authors being an employee of the private-sector funder (Pfizer). There needs to be a very careful explanation of how any associated conflicts of interest were managed in respect of this research.

This study was funded by Pfizer Japan, Inc. and Naoko Izumi, who is one of the co-authors, is an employee of Pfizer Japan Inc. and has contributed to preparation of the study design, data analysis, decision to publish and preparation of the manuscript. However, Pfizer Japan Inc. did not provide support in the form of salaries for authors other than Naoko Izumi. Pfizer is a marketing authorization holder of fesoterodine, which is an anti-cholinergic medicine for overactive bladder and pediatric neurogenic bladder. This information does not alter our adherence to PLOS ONE policies on sharing data and materials.

3.The authors should be very clear about when they are referring to "xerostomia" (the symptoms of dry mouth) and when they are referring to low salivary flow (or salivary gland hypofunction, SGH). The subjective and objective aspects of dry mouth do not necessarily coincide, and the authors' choice of terms should reflect that dichotomy. People with an unstimulated flow rate of <0.1 mL/min do not necessarily have xerostomia, but they definitely have SGH.

Thank you for your comments. As pointed out by the reviewer, subjective feeling and objective evaluation do not match. In our study, because all subjects’ unstimulated flow rate was <0.1 mL/min, all subjects had hyposalivation. Regarding medication-induced xerostomia, the mechanism of hyposalivation is clear with some medications, while that for others is not clear. Therefore, in this article, we defined xerostomia as the subjective feeling of dry mouth, and analyzed its characteristics and treatment effects.

4.The study design is not clear, but I think I have worked it out - it could be described as a retrospective cohort study of a large clinical convenience sample, but there is an actual intervention, so it's not strictly observational (and therefor enot a cohort study as such). I think it is a case series analysis a large clinical convenience sample for which there was an intervention (or interventions). The authors need to be very clear about their design, because the reader's understanding of the findings is very much dependent on the design being explicit.

We thank the reviewer for the advice. We have added the following text to the Methods section in the revised manuscript: 

“This was a case series analysis of a large clinical convenience sample for which there was an intervention.”(line 77).

5.Always hyphenate "side-effect". It is a compound word.

In accordance with the reviewer’s comment, we have hyphenated this word as suggested.

6. Headings and subheadings are huge - do we really need them that big?

This is PLOS ONE’s style.

Section comments

Title

This needs to better reflect the study design and the research question, and the term "xerostomia" should be replaced by "dry mouth". 

We thank the reviewer for the advice. We have changed the title to ‘Characteristics of medication-induced xerostomia and effect of treatment’. Regarding the term xerostomia, it is defined as a subjective feeling of dry mouth. Therefore, xerostomia and dry mouth have the same meaning. The term “xerostomia” is commonly used in the literature as well as “dry mouth.”

Abstract

Will need rewriting anyway, but some comments follow.

The logic of the first two sentences escapes me.

We have changed the sentence as follows: 

“Side-effects of medications cause xerostomia. There have been cases where a medication has been discontinued owing to its severe side-effects.” (lines 22-23).

Use the past tense in describing findings. Do not rely on P vlues - put some actual data in the Results paragraph. Your study should not be a P value hunt - see Amrhein et al, Nature 2019; 567: 305-307.

We have changed this text to the past tense for describing findings. We have also added the actual data.

Conclusion - that phrase "…can prevent decreased drug compliance for the primary disease" is also a mystery - what are you saying? By what logic?

We changed the sentences as follows;

“If xerostomia due to side-effects of medications can be improved by treatment, it will greatly contribute to the QOL of patients with xerogenic medications and may reduce the number of patients who discontinue medications.” (lines 42-44).

Introduction

Overall, this should make a more compelling case for conducting the study.

Paragraph 1

1. Sentence 1 - change "decreased" to "low". For a better and more recent estimation of prevalence rates, see Agostini BA et al. How common is dry mouth? Systematic review and meta-regression analysis of prevalence estimates. Brazilian Dental Journal 29: 606-618 (2018). 

We have changed “decreased” to “low”, and we have added the prevalence of xerostomia (lines 46-47).

Last sentence - that is an unsupported sweeping statement - where is the evidence for that assertion?

We have added a reference (line54).

Paragraph 2

Sentence 2 - medication-induced dry mouth accounts for well over 95% of cases of dry mouth - see the recent US Surgeon General's Report on Oral Health.

We thank the reviewer for the advice. We are referring to patients who are specialized xerostomia outpatients. We have added a reference (recent US Surgeon General's Report on Oral Health) to the revised manuscript (line 67).

Sentence 3 - That Sreebny list is far too inclusive, and you should avoid using those specific numbers.

We agreed to the reviewer’s comment. We have deleted the relevant sentences.

Paragraph 3

See the Surgeon General report for a good overview of the therapeutic approaches for treating dry mouth - there is a very useful Table there. Last sentence of para 3 - change the awful "regarding" to "on".

In accordance with the reviewer’s advice, we have added a reference (line 67). We have also changed “regarding” to “on” in the revised manuscript (line 69).

Paragraph 4

Sentence 1 - "…and eventual imrpovement of the primary disease" - really? That's drawing a very long bow. How might that work? Next sentence - change "Therefore" to "Accordingly". And specify the research question. The way the last sentence is worded is far too loose and woolly, and the reader has no idea of what the study is about.

We have changed the following sentences:

“If xerostomia due to side-effects of medications can be improved by treatment, it will greatly contribute to the QOL of patients with xerogenic medications and may reduce the number of patients who discontinue medications. Accordingly, this study aimed to analyze the characteristics of patients with medication-induced xerostomia and the effect of treatment of xerostomia” (lines 71-74).

Methods

Lines 93-94 - patients were not identified; the drugs were. Rewrite that sentence accordingly.

We thank the reviewer for the comment. We have deleted the phrase “Patients taking” (line 94).

Lines 105-107 - express those as mL/min, not per 15 min.

We have changed the unit to mL/min．

Lines 111 to 113 - how do you know they "developed xerostomia"? Or did they "have" the condition?

 We changed developed to onset. We defined psychological stress-induced xerostomia when psychological stress events and onset of xerostomia were occurred simultaneously (line113).

Line 121 - "educated regarding" is poor wording - they were "made aware of" it, or taught how to do it - something like that is better.

We have changed “educated” to “instructed” (line 121).

Lines 126-7 - what herbal medicines? How given? When? Why? No reference - just a throwaway comment is given.

We have added the name of the herbal medicines and a reference (lines 126-127).

Line 135 - how exactly was xerostomia assessed? What was asked, and what were the response options?

We have changed the sentence as follows: 

“Changes in the severity of xerostomia were assessed as three levels;”(lines 135-136).

Line 137 should be reworded.

We have changed the sentence as follows: 

“If follow-up did not continue until 6 months, the data at the time of completion were treated as data at 6 months.”(lines 138-139).

Statistical analysis section - concentrate less onf the test stistics (such as chi-square tests) and more on the procedures - such as "cross-tabulations", etc etc.

We have changed the following sentence: “Cross-tabulations were performed with the chi-square test or Fisher’s exact test for categorical variables, and the Cochran–Armitage test was used for categorical variables with three or more categories.” (lines 143-145).

Results

Line 159 makes no sense at all. Line 166 also has problems - "all types" - really? Do you mean "any type"?

We have changed the following sentence:

“We included 490 of the 1,378 patients in this study on the basis of the selection criteria.” (line 165)

In addition, as you pointed, we have changed “all types” to “any type” (line 172).

Table 1 is just a list. What not make it more informative for readers by cross-tabulating by sex or age group? And no Table should continue onto a second page. Think about reducing your use of gridlines - they actually distract the eye from the data.

This is the first evidence of the actual situation of patients with medication-induced xerostomia in Japan. Therefore, we consider that we need to show the patients’ background in detail.

In the Results text, do not repeat Table data - the reader can see the data in the Tables; your job is to draw his/her attention to the important parts and features. Have one paragraph of Results text per Table - this helps the reader to navigate your paper - and introduce each Table at the beginning of its paragraph, not at the end. Concentrate on what the data show, NOT on the P values - they are far less important.

We thank the reviewer for the advice. We have rewritten the results, and moved the tables to the beginning of the paragraph.

Avoid using the term "compared to" when making comparisons - use 'than' - for example, in Line 227, replace "increased compared to baseline" with "was higher than at baseline'.

We have changed "increased compared to baseline" to "was higher than at baseline" (line 228).

Table 3 would be enhanced by presenting effect sizes (the difference in mean score divided by the SD of the baseline score, and prsented to 1 decimal place. For example, that for those <65 years is 1.0, which is a large effect. And you will be able to demonstrate that the ES for the Sjogren's patients was lower (0.4) than for those with medication-induced xerostomia (1.0) - that is important information, and much more useful and informative than the P values - which you could indicate with a footnote anyway.

We thank the reviewer for the advice. We have added the effective score to Table 3.

Discussion

This will need rewriting anyway - in its current form, it is too long and discursive. In the Discussion and conclusion, use the term 'findings' rather than 'results'. See Docherty and Smith, BMJ 1999; 318: 1224-5 for how to structure a Discussion section. It is a useful structure. As a general rule, the first paragraph of the Discussion should briefly reiterate what the study did and what it showed. The second paragraph should address the weaknesses of the study design and measures, etc. The paragraphs which follow should then discuss how the findings support or refute the current literature. The final paragraph should tie it all together - so what? Where next? What are the implications for practice?

We have changed the term “results” to “findings” as suggested. We have also rewritten the Discussion.

The current conclusion is not useful - what's the take-home message from this study?

We have changed the conclusion as follows: 

“The doctors who prescribe xerogenic medications should attempt to manage xerostomia. This would prevent patients with xerostomia from no longer taking medicine for the primary disease.” (line 323).

Reference 23 - is that legit?

We have changed the reference to number 26.

Reviewer #2: 

Line 63 - You better explain what OAB stands for. Does it refer to overreactive bladder?

We thank the reviewer for the advice. We have changed “OAB” to “overactive bladder” (line 63).

Line 72 - You better explain what QOL stands for. Does it refer to quality of life?

We had already defined QOL on line 51 of the manuscript.

In the results section I would suggest you to diplay the results in tables only as the majority of them were explained in the text and repeated on the tables. You could have a small paragraph describing a little about what will be displayed without mentioning the figures. I would also suggest you to re-organize table 1 as data are not clear. For instance, I took some time to understand that data presented as 70 on line 2, stands for the median age, and on the next line the figure stands for the size of the sample >= 65 years old. Its seems confuse!

On the headline of the columns write to what the figure stands for and in brackets you could write frequency in %. You better verify the total percentage for medical history.

We thank the reviewer for the advice. We have modified Table 1 in accordance with this comment and the other reviewer’s comments. Because the medical history was duplicated, the total percentage was > 100.

Line 183 - What kind of Japanese Herbal medicine was used? Do they all have the same mechanism of action?

We have added the name of the herbal medicines and a reference. These medications facilitate the secretion of saliva.

Did the use of anticholinergic drugs present any type of pharmacological interaction with the other drugs in use by patients? Did you have this data on patient´s record? Nothing was mentioned on the text about it.

Unfortunately, we could not determine the interaction of the medications. We have added this issue as a limitation to the revised manuscript (line 308).

---

## [Editor Report · Decision Letter 1]

26 Dec 2022

Characteristics of medication-induced xerostomia and effect of treatment

PONE-D-22-09854R1

Dear Dr. Ito,

We’re pleased to inform you that your manuscript has been judged scientifically suitable for publication and will be formally accepted for publication once it meets all outstanding technical requirements.

Kind regards,

Sompop Bencharit, DDS, MS, PhD, FACP

Academic Editor

PLOS ONE

Additional Editor Comments (optional):

Thank you for the revision.

Reviewers' comments:

The authors have sufficiently addressed all comments.

---

## [Editor Report · Acceptance letter]

3 Jan 2023

PONE-D-22-09854R1 

Characteristics of medication-induced xerostomia and effect of treatment 

Dear Dr. Ito:

I'm pleased to inform you that your manuscript has been deemed suitable for publication in PLOS ONE. Congratulations! Your manuscript is now with our production department. 

Kind regards, 

on behalf of

Dr. Sompop Bencharit 

Academic Editor

PLOS ONE